# The Structure of Social Support in Confidant Networks: Implications for Depression

**DOI:** 10.3390/ijerph18168388

**Published:** 2021-08-08

**Authors:** Liyuan Wang, Lindsay E. Young, Lynn C. Miller

**Affiliations:** Annenberg School for Communication and Journalism, University of Southern California, Los Angeles, CA 90007, USA; lindsay.young@usc.edu (L.E.Y.); lmiller@usc.edu (L.C.M.)

**Keywords:** egonet, network constraint, clinically depressed, network interventions

## Abstract

Social support differs for depressed and non-depressed individuals. However, the structural features of social supports, as represented via social networks and how they are related to depression, and its mitigation, are unclear. Here, we examine associations between personal support network structures and self-reports of depression and depression mitigation behaviors. Cross-sectional data were collected from participants (*n* = 1002 adults) recruited from a research volunteer website. Personal support networks were elicited by asking participants to nominate up to six people (i.e., confidants) that they talk to about interpersonal problems (e.g., unpleasant social encounters) and to indicate who knows whom among their confidants. Results show that the confidant networks of depressed and non-depressed participants did not differ in network size or in constraint—i.e., the degree to which confidants’ ties overlap with the ties of the participant. However, depressed participants’ confidants had significantly fewer average ties with one another (mean degree). Irrespective of depression diagnosis, lower network constraint and size predicted greater engagement in depression mitigation behaviors. That is, having relatively large confidant networks within which one can freely navigate one’s personal information can contribute to improvement in depressive outcomes. Implications are further discussed in the discussion section.

## 1. Introduction

In the United States, there are over 17.3 million adults (7.1% of the adult population) who have experienced a major depressive episode in the past 12 months [1], making depression one of today’s most prevalent chronic adverse health outcomes [2]. Using almost any metric, depression is both debilitating and costly. More than a psychological burden, depression has numerous long-term health consequences, including heart disease, injuries, inflammation, and lowered immunity functioning, and has been linked to suicide and suicide ideation [3,4]. The economic burden of depression resulting from disability, morbidity, and mortality is estimated to be $USD 210 billion annually [5].Further, the CDC reports that one-year after COVID-19, the percentage of adults reporting recent symptoms of anxiety and depressive disorder increased from 36.4% to 41.55% [6].

Therefore, after a time of isolation and quarantine, it is especially important to understand what (e.g., which features of the social environment) might help mitigate depression and, therefore, should be targets of future intervention trials. For clinicians, depression is typically defined as a persistent intra-psychic disorder in which individuals feel sad and hopeless and derive little pleasure or have little interest in activities, but depression is often influenced by the social contexts in which an individual is embedded [7,8]. To study these social networks, researchers often measure personal (or ego) networks: these are the self-identified set of social relationships immediately surrounding a focal individual who provide support and protect that focal individual from becoming depressed or help him or her more effectively cope with the depression symptoms [9]. Although the positive relationship between social support and protection from depression is well-documented [9,10,11,12,13,14,15], there is a significant research gap: it is unclear how pathways of support are organized (i.e., the structure of personal support networks) for depressed and non-depressed individuals and how those structural features impact an individual’s engagement in critical depression mitigation behaviors. In one of the few published findings in this area, Hall and colleagues [12] found that among seniors (i.e., 62 and older) the size of the ego’s self-reported network and its mean degree (number of observed links divided by potential links among network members) promoted open communication, which in turn can mitigate depression. These are promising findings but have three limitations. First, these networks are not specific to the types of topics (e.g., important or intimate matters) expected to be central in mitigating depression (ref). Second, it was unclear if these findings would generalize to other age groups. Third, the structural features examined of these networks were limited.

Indeed, social support networks and their features can be understood in terms of the role of a specific type of personal support network called a confidant network. A confidant network consists of the ego and the ego’s strong-tie relationships (i.e., confidants) to which the ego turns to talk about important matters [12,16,17]. Through their companionship and support, confidants are integral to an individual’s ability to avoid and/or attenuate feelings of distress and depression. What, however, are the network structural features of social support (or confidant) networks that may effectively mitigate depression? The answer to this question remains unclear.

To fill this gap by drawing on cross-sectional confidant network data collected from 1002 adults, we investigate: (1) the degree to which the confidant networks of depressed and non-depressed individuals are structurally and compositionally different, and (2) the degree to which three structural features of confidant networks—i.e., the number of confidants (network size), average number of ties of a network member (mean degree), and degree to which confidants are tied to other confidants (constraint)—are associated with engagement in depression mitigation behaviors (e.g., activities that promote personal mental health benefit). Answering these questions has implications not only for how we theorize the relational infrastructure that underpins social support in both depressed and non-depressed populations but also for depression mitigation interventions.

## 2. Background

### 2.1. Confidant Networks and Psychological Well-Being

Social networks are patterns of human relationships—generally defined by who knows whom or who communicates with whom—through which resources and benefits are transmitted and obtained [18]. Participating in social networks is viewed by many as foundational to human existence [2,10,19] and, therefore, has been theorized as an essential component of psychological well-being [20,21,22]. In this vein, personal networks are of particular value. Personal networks consist of a focal individual (i.e., ego) and the people with whom the ego is directly connected to (i.e., alters) along with the relationships that exist among the alters [18]. As the most immediate set of social contacts on which a focal individual can rely, personal networks have garnered attention for their direct and mediating relationships with psychological well-being and, relatedly, their role in helping individuals cope with routine problems, crises, and burdens through their facilitation of social support [9,23,24,25,26].

Personal networks can consist of multiple layers of social interaction that differ in intimacy, ranging from an individual’s participation in various communities, to their interpersonal interactions, to their closest “core” relationships [13,17]. It is this most inner (core) layer that is thought to have the strongest association with lower levels of psychological distress and depression [13]. Accordingly, this study focuses on one particular manifestation of this inner layer of personal networks—the confidant network. Confidant networks are composed of the network associates with whom a focal individual communicates regularly about matters of importance [17] and are typically composed of strong-tie relationships. As sources of companionship and emotional and instrumental support [27], confidants play a critical role in an individual’s mental health and their resilience to life stresses [12].

Although studies have shown that the specific “important matter” that an individual wants to discuss informs which confidants they activate and, relatedly, the nature of the support they seek and receive [28] one can argue that confidants have a more generalized function, too. Irrespective of the topic of conversation, confidants can have positive influences on an individual’s mental state through the self-focused reassurance and feedback they provide that, in turn, may help the focal individual engage in behaviors that promote personal benefit [12,29]. Indeed, clinicians have long sought psychotherapeutic means that can relate to depression mitigation outcomes through the depression activation system [30,31,32], which consists of (1) increasing engagement in adaptive behaviors, (2) avoiding engagement in maladaptive behaviors and problematic problem-solving behaviors that maintain or exacerbate depression. Then perhaps social support structures can shed light, for behavioral therapies, on new ways to increase these depression mitigation behaviors.

### 2.2. Current Study: The Structure of Social Support and Depression Mitigation

Social support has a long history as the key to depression mitigation [33,34]. Social support, in this study, defined as the extent to which an ego perceives that friends, family members, and others provide various types of psychological and other support when needed, consistently predicts physical and mental health outcomes [35]. Indeed, researchers have consistently found that a social support network can usually make one more resilient towards daily stressors, therefore, protecting one from depression. From a structural perspective, the amount and quality of perceived social support that a focal individual receives is, in part, derivative of the arrangement of social ties around them. Accordingly [20], studies examining the structural characteristics of personal support networks in relation to psychological well-being and depression have traditionally crystalized around the role of personal network size. The size of a personal confidant network (i.e., the number of strong tie contacts that a person can potentially communicate with and receive support from [12], is perhaps the most fundamental and intuitive structural indicator of how much social support an individual can potentially receive [12]. The essential takeaway from this body of work is that smaller social networks and fewer close relationships are routinely linked to depressive symptoms [24,36].

Several mechanisms by which personal network size influences psychological health have been posited. In one formulation, the degree of an individual’s social connectedness produces positive psychological states such as a sense of purpose, belonging, and security, as well as feelings of self-worth and self-competence. These positive states of mind, in turn, may benefit downstream mental health through an increased motivation to engage in self-care (e.g., engaging in activities that offer personal benefit) or by increasing an individual’s cognitive capability to handle stress [24,37].

Structural theories of social capital [13] suggest another route through which network size positively impacts psychological well-being—i.e., through the resources and opportunities that supporters provide. From this perspective, the size of one’s support network is indicative of the number of resources one has available to prevent and/or cope with depression and the adverse life events that can cause or exacerbate depression downstream [13], for example, resources in the form of informal mental health care and advice. Further, individuals with larger support networks may have more opportunities to engage in beneficial social activities that mitigate or distract from feelings of depression. Taken together, we expect a positive relationship between the size of an individual’s confidant network and their engagement in depression mitigation behaviors. Indeed, this consistent pattern of empirical results, which is also consistent with relevant theories in this domain, allows us to formulate the following hypothesis.

**Hypothesis** **1.**
*Confidant network size (i.e., number of confidants) is positively associated with engagement in depression mitigation behaviors.*


As previously noted, personal networks also encapsulate patterns of ties among a focal individual’s alters. However, largely absent from the foundational literature on the relationship between social support and depression is how the presence (or absence) of observed relationships among the supporters themselves impacts a focal individual’s psychological well-being. Cohesion, indicating the level of connectedness among network members, has been used to indicate the level of social support within a given network [38]. Earlier, we referred to mean degree and constraint: both are, albeit different, measures of cohesion. Drawing on theories of social capital [27,39,40], two archetypal formulations of the personal value accrued from structural embeddedness regarding cohesion emerge. The first is dense personal networks, within which alters have mutual ties with each other (represented by high mean degree, Figure 1B, relative to other networks (e.g., Figure 1A)), that are optimal for social support (i.e., bonding social capital). The second is radial personal networks, within which alters connect to the ego but not with each other (represented by low network constraint, Figure 1), that are suited for accessing novel information and/or advice (i.e., bridging social capital [41,42]).

Depending on one’s perspective, both types of personal network structures could enable engagement in depression mitigation behaviors. On the one hand, when cohesiveness of a network is evaluated as the extent to which ego is embedded in a web of social support that includes a greater number of connections among confidants (density, measured with a high average mean degree), high cohesiveness (dense) networks are likely to yield greater mutual trust and accountability among network members and provide the focal individual with a stronger sense of belongingness and emotional wellbeing [13]. Further, the concentrated flows of information that more cohesive networks produce may reinforce supportive messaging to a focal individual, thereby generating greater self-worth and a greater desire to engage in activities that promote personal benefit.

On the other hand, cohesiveness in a network is also evaluated in terms of the extent to which the ego’s actions and perceptions are controlled by the close connection among alters. For some, the cohesiveness of a confidant network may actually be constraining, as it limits the amount of new information or advice that a focal individual can receive from their confidants, as the ego is bonded with other alters [18,19]. That is, when a confidant network is less constrained (i.e., when confidants have fewer ties that overlap with the focal individual’s ties), the ego is the bridge that connects other alters, and the focal individual may be positioned to have greater command over their own support system, enabling her/him to seek support from confidants in a more intentional and strategic fashion.

Given different evidence-based perspectives on the value of bonding (more constraint) versus bridging (less constraint) social structures and our own speculations about its potential positive and negative roles in depression mitigation, there is no clear-cut directional hypothesis here. Instead, we ask the following research question.

**RQ:** 
*How will confidant network cohesion be related to a focal individual’s engagement in depression mitigation behaviors?*


## 3. Methods

### 3.1. Study Design and Sample

This study is of a survey design. Data were collected in August 2019 and in September 2019 as part of an online survey study designed to investigate the relationship between depression and features of personal networks. Participants were recruited via a volunteering website developed with funding from the National Institute of Health for clinical trials: Researchmatch.org. This website (ResearchMatch.org accessed on 7 June 2020 with a national web-based registry of more than 150,000 patients looking for new treatment of their disease or condition, allows researchers to access volunteers with existing health conditions who are considering participating in research studies or clinical trials [43,44,45]. ResearchMatch.org caters to the public desire to find recruiting research trials for their conditions and the need for researchers to conduct research that might produce new therapies and treatments for targeted populations. This website offers a registry that is freely available for anyone to use and to enter their goals for trial participation, medical condition, and geographical information. ResearchMatch.org could be an ideal site for our research because depression and anxiety were the top conditions that were searched by its volunteers [46]. By the year 2020, Researchmatch.org documented 150,364 registered volunteers, 870 projects, and 491 academic publications (for details, see Researchmatch.org). Candidate participants were eligible if they were (1) 18 years of age or older, and self-reported either (2) a diagnosis of clinical depression or (3) no diagnosis of any mental health disorder. Participants’ depression diagnoses were screened both by Researchmatch.org and their subsequent self-reports upon their agreement to participate in our study. That is, participants needed to establish their own profile on Researchmatch.org in order to be contacted by researchers. In this study, we use the filter provided by the website and send inquiries to participants regarding their willingness to participate in our study. We set the filter to include participants who documented a recent diagnosis of depression. This information was also later manually verified by researchers in terms of the self-reported medication of participants (that is, did the participant report a medication that is used to treat depression). We excluded participants who self-reported other mental health conditions (e.g., schizophrenia) using the filter provided by researchmatch.org.

After initial contact was made, we further screened to see if the participant’s diagnosis was current. Participants were excluded if they (1) did not specify a date of diagnosis, or (2) did not specify which depression disorders they had. Power analysis showed that to detect a small level of difference, at least 63 people per group was needed. Our sampling procedures resulted in a sample of 1002 adults (620 non-depressed and 382 clinically depressed) with no missing data, ensuring sufficient power to detect a difference. Upon completion of the online survey, participants were entered into a lottery to win a $USD 50 gift card. The study was approved by the Institutional Review Board of authors’ research institution, and all participants provided consent.

### 3.2. Measures

#### 3.2.1. Depression and Depression Mitigation

The degree to which participants engaged in depression mitigation activities was measured using items from the Behavioral Activation for Depression Scale (BADS) [31,32]. The BADS scale consists of 9 items that represent weekly behavior thought to underlie depression. Each item on the scale is measured on a seven-point scale ranging from 1 (not like me at all) to 7 (a lot like me). This scale has been widely used to (1) track depressed patients’ engagement in behaviors that may improve/exacerbate their depressive symptoms, (2) monitor their responsiveness in terms of Depression Behavioral Activation Therapies, and (3) to evaluate their overall depressive symptoms in the past week [43,44]. The scale consists of two dimensions. The first dimension encompasses four items that measure avoidance or rumination behaviors, which are associated with depression xQA outcomes (e.g., “I kept trying to think of ways to solve a problem but never tried any of the solutions” or “I spent a long time thinking over and over about my problems”). The other dimension has five items that measure activation behaviors, which are activities that promote positive personal benefits (e.g., “I did things that were enjoyable” or “I am content with the amount and types of things I did”). As the focus of this study is about one’s ability to prevent the onset of depression and/or attenuate its symptoms, we use the latter five behavior activation items to form a composite variable representing overall depression mitigation engagement (Cronbach’s α = 0.85). That is, we use the additive value of the five items divided by five. A score of 7 means the highest possible score of depression mitigation behavior and a score of 1 means the lowest. Each of the five scale items and their means and standard deviations are shown in Table 1. It is worth pointing out that counter-intuitively, clinically depressed participants reported higher levels of depression mitigation behaviors than those who are not clinically depressed (implication discussed in later sections). While this pattern is unexpected, it does not affect the main goal of the study.

#### 3.2.2. Generating Personal Confidant Networks

A central component of the online survey assessment was the enumeration and description of the personal support networks of participants. To this end, we used a name generator—i.e., an instrument used to identify the individuals (or alters) who are in a focal individual’s (or ego’s) personal network [17]. Name generators are the workhorses of personal network data collection and have been used to capture a variety of personal networks including social support networks [17] and core discussion networks [17,45]. The confidant name generator elicited up to six confidants in the past 3 months (including a romantic partner if one was reported by the participant) using the prompt, “Please name the people to whom you talk about your interpersonal problems” (e.g., unpleasant social encounters, upsetting co-workers, relationship trouble, etc.). Once confidants were named, participants were also asked to indicate who knew whom among the nominated confidants. Egonet ties were formed based on whether members of the egonet knew one another. They were also asked demographic questions about each confidant, including their gender and their relationship to the participant (e.g., family, friend, romantic partner). The choice of 5 to 6 nominations have been widely adopted by self-reported network measures [46]. Generally, a nomination of 5 to 6 is the recommended “balance” to avoid the redundancy of having to name too many nominations (over-reporting those who are not confidants) and the failure to capture the actual peer social network structure by naming too few links [47]. These data are then used to graph the personal confidant network with the participant (or ego) and a series of confidants (or alters) that are connected via confidant ties from the ego to each alter and generic social ties between the alters.

#### 3.2.3. Measures of Network Structure

Structural characteristics of confidant networks include network size, mean degree, and constraint. Network size represents the number of confidants nominated by a participant (up to 6), which is the total number of nodes within an egonet. To account for the cohesiveness of confidant networks, we use two measures. The first measure is mean degree, which represents the average number of ties (connections between each node) of a confidant, excluding the participant, indicating the distribution of alter-to-alter ties in the confidant network [41]. The second cohesive measure is called constraint [39]. Constraint is a summary measure of the degree to which a participant’s connections are to confidants who are themselves connected to one another [48]. Cohesive networks are indeed a common structural signature of social support, however, the idea behind constraint is to underscore cohesiveness’s disadvantages, revealing how having more confidant ties can in fact restrict a participant’s freedom of action if those confidants are also connected to one another. Although both mean degree and constraint represent connectedness within the network, there is an important distinction: mean degree is a measure of cohesiveness among alters only, while constraint captures the degree to which the ego is affected by the connectedness among alters [18].

#### 3.2.4. Measures of Network Composition

Although not our primary focus in this study, we also examine the relationships between compositional features (i.e., the proportion of characteristics across members in a network) of confidant networks and depression diagnosis and engagement in depression mitigation behaviors. Specifically, we include measures representing the proportion of kin and the diversity of gender in each participant’s confidant network. Proportion kin is the ratio of network members who are also family. The diversity of gender index (or the index of qualitative variation) represents the mix of men and women in the network with a value of 0 meaning all network members are one sex and a value of 1 indicating an equal mix of men and women [49]. In short, the index was calculated by using the sum of the proportions of each gender, the number of gender categories (here male and female), and the sample size (for details of the IQV formula, refer to [49].

### 3.3. Statistical Analysis

We first conducted a series of tests to determine whether there were systematic patterns in missing data. Results of those tests indicated that there was no systematic missingness. Therefore, we used the default pairwise deletion in SPSS to deal with the missing data. Using a series of t-tests, we assessed differences between clinically depressed and non-depressed individuals in terms of the structure of their personal confidant network. It is worth pointing out that while this study conducted t-tests, it is not a controlled experiment. Therefore, we did not conduct any other procedures to control or manipulate missing data except pairwise deletion. We then used hierarchical linear regression to estimate associations between features of network structure and engagement in depression mitigation behavior. To avoid multicollinearity, each of the three measures of network structure was examined separately, while adjusting for measures of network composition and clinical diagnosis. Specifically, three adjusted hierarchical linear regression models were estimated. Hierarchical linear regression should not be confused with hierarchical linear modelling, a type of analysis appropriate for multi-level data structures. Hierarchical linear regression is used when researchers want to see whether the addition of a variable (or set of variables) significantly improves model performance. As such, covariates are added to the model in separate steps (or “blocks”) [50]. Each hierarchical linear regression model featured here begins with a first block of covariates that have been identified in previous work as being related to depression mitigation, including network composition variables (proportion kin and gender diversity) and current depression diagnosis. The second block adds to the first block the focal measure of network structure (i.e., either network size, mean degree (or edge count), or constraint). All network measures were computed via the Statnet and igraph packages in R. All other descriptive and statistical analyses were performed using SPSS v. 26.

## 4. Results

### 4.1. Participant Characteristics

A summary of sociodemographic characteristics of the sample, stratified by depression status, is shown in Table 2. Results show that depressed and non-depressed participants were comparable in demographics, with majorities in both subsamples being White/Non-Hispanic, earning less than $USD 60,000 per year, and female. Participants in both samples were on average 37.4 years of age (SD = 12.79). There are also no significant differences between the two samples regarding age, years of education, and race/ethnicity. In preliminary analyses (not shown here), we also entered age, education, race/ethnicity, income, religious/political affiliation as covariates; none of them were significantly related to depression mitigation behaviors. Therefore, none of those variables were included in the featured analysis.

Given the gender bias in our samples (82% of depressed participants and 79% of non-depressed participants were female) and known associations between gender and depression, we performed a series of t-tests (not shown) to compare network measures between females and males. There were no significant differences by gender regarding the network structure measures, however, female participants reported greater gender diversity in their personal confidant networks (*p* < 0.01). As such, interpretation of our findings should be made with this caveat in mind.

### 4.2. Confidant Network Characteristics

Irrespective of depression status, participants reported on average *M* = 4.17 confidants (SD = 1.68) with a range of 0 to 6. Both samples named a similar proportion of family members in their network (30% in the depressed sample and 29% in the non-depressed sample). As expected, depressed participants reported more connections with therapists, 46 out of 390 depressed participants reported having therapists in their network, while only 17 out of 686 non-depressed participants named a therapist as a confidant.

It is worth noting that 37 participants (*n* = 29 in the depressed sample and *n* = 8 in the non-depressed sample) reported no confidants. Some participants nominated pets or gods (*n* = 7 in the depressed sample, *n* = 20 in the non-depressed sample) as one of their confidants. Although we recognize that pets and spiritual figures play important roles in the lives of these participants, we opted to exclude these participants from the analysis.

To understand how this exclusion decision might affect our analysis, we compared differences in depression mitigation behavior between those who did not nominate a confidant (or did not name a human confidant) and those who nominated at least one eligible confidant. Not assuming equal variances, independent t-tests showed significantly lower depression mitigation engagement among participants who did not have a confidant (*M* = 3.84, *SD* = 0.17) than those who did have at least one confidant (*M* = 4.35, SD = 0.04) (*p* < 0.01, *t* (1052) = 3.21, 95% CI:(−0.68, −1.65), Cohen’s *d* = 4.13).

### 4.3. Differences in Network Structure: Depressed Versus Non-Depressed Samples

Table 3 provides the raw scores of each network measure separately for the clinically depressed and non-clinically depressed samples. Results indicate that while depressed participants nominated fewer confidants on average (*M* = 4.15, SD = 1.71) than non-depressed participants (*M* = 4.19, SD = 1.63), this difference is not significant (*t* = −0.31, *p* > 0.50). There were also no significant differences in network constraint between depressed (*M* = 0.61, SD = 0.15) and non-depressed (*M* = 0.61, SD = 0.14) participants. Finally, there were also no significant differences in the family/kin composition of depressed (*M* = 0.29, SD = 0.38) and non-depressed (*M* = 0.29, SD = 0.34) participants. We only found significant differences between the confidant networks of depressed and non-depressed participants in terms of mean degree, the confidants of depressed participants had significantly fewer ties with the other confidants in their networks (*M* = 0.74, SD = 1.12) than the confidants of non-depressed participants (*M* = 1.95, SD = 1.12) (*p* < 0.01). Other measures (i.e., network size, constraint) were not significantly different.

### 4.4. Relationships between Network Structure and Depression Mitigation Behaviors

Among the 1002 participants, 214 (21%) reported fewer than three confidants. As some of the network structure variables cannot be computed on personal networks with fewer than three alters (e.g., personal network constraint), we opted to restrict our analytic sample for the second portion of the analysis to only participants who nominated three or more confidants. In a pre-test (not shown here), we compared the differences in depression mitigation engagement between those who nominated fewer than three confidants and those who named three confidants or more. Not assuming equal variances, independent t-tests showed no significant differences.

Results of the hierarchical linear regression models are shown in Table 4, Table 5 and Table 6. Network size predicted depression mitigation engagement (***B*** = 0.11, SE = 0.04, *p* < 0.01), such that larger network size was associated with more depression mitigating behaviors. This supports our H1, which predicts the positive associations between network size and depression mitigation behaviors. While mean degree was not significantly related to actively engaging in depression mitigation behaviors (***B*** = −0.05, SE = 0.07, *p* > 0.05), network constraint is related to depression mitigation behaviors (***B*** = −1.15, SE = 0.48, *p* < 0.05), such that having a more constrained confidant network was negatively associated with more depression mitigation behaviors, providing answers to our research question on the role of constraint in mitigating depression behaviors (see detailed discussion of this result in the discussion section). Furthermore, all models were adjusted for depression status, and in all cases, having experienced clinical depression did not significantly predict depression mitigation engagement. That is, clinically depressed individuals neither uniformly differed from non-depressed individuals in the degree to which they engaged in behaviors to mitigate their depression when network structure factors were accounted for.

## 5. Discussion

This study, for the first time, investigated associations between confidant network structures and depression mitigation behaviors in an online general sample of clinically depressed and non-depressed adults across the age continuum. We also compared the confidant networks of clinically depressed individuals and non-depressed individuals. Although simply having a confidant network was a significant predictor of one’s engagement in depression mitigation behaviors, as our preliminary analysis showed, we also learned that, irrespective of depression status, specific network structures—namely, network size and constraint—played what could be meaningful roles in enabling depression mitigation engagement, at least among those who had a confidant network. These results remind us of the importance of considering an individual’s social embeddedness when accounting for the factors that facilitate or impede their ability to cope with life stresses. Therefore, the featured analysis adds to existing empirical research on the role of interpersonal factors as main predictors of mental health outcomes. Specifically, we showed that in addition to our existing knowledge about the role of network size (i.e., the number of confidants that a person has access to) in mitigating depression, it is important to also consider the relationships among confidants (e.g., the possible value of low constraint) in order to more fully understand the effects of network structure on depression mitigation. These findings suggest a possible intervention target, namely, adding a network member (e.g., a therapist), who does not have a relationship with the ego’s other social network members, might enhance ego’s depression mitigation behaviors. We discuss the findings for constraint in this regard in more detail below.

Our findings have implications for how we think about depression and its mitigation. First, we did not find a statistically significant difference between depressed and non-depressed individuals regarding network size in our sample. However, this result is not inconsistent with prior work that has identified associations between social isolation (lack of social connections) and depression [51]. Keep in mind that in this analysis, we only included participants who nominated at least three confidants—those are participants that may already have sufficient levels of social support. Also, from those participants, we found network size played a significant role in mitigating depression, which is consistent with previous studies that focus on the size of networks in protecting one against depression [2,52]. Our results on network size, therefore, have positive implications: when in a time of need, clinically depressed people may have a sufficient confidant network upon which to rely, at least at the onset of their depression. The number of confidants one has in one’s social network matters in helping one seek behaviors that can contribute to the improvement of one’s depression. Indeed, it is necessary for researchers to look into ways that can help clinically depressed individuals in maintaining or expanding their confidant networks for the sake of improving their depressive symptoms.

Second, in our comparison of the network structure of depressed and non-depressed participants, we found significant differences in mean degree such that the confidants of non-clinically depressed individuals were, on average, more connected than those who are clinically depressed (shown through mean degree). This is not unexpected when evaluating a confidant network. Additionally, as reported in the previous section, only 2% (17 out of 686) of non-depressed participants reported having therapists/mental health professionals outside of their regular confidant networks. Indeed, non-depressed individuals’ networks can be more “natural” given that non-depressed people reported highly connected clusters of individuals as their confidants. However, our results showed that this type of cohesive structure—indicated by the average of mutual ties shared among confidants—does not necessarily enable depression mitigation behaviors. This is also not unexpected because having more confidants who know one another does not necessarily motivate behaviors to manage or mitigate depression. Having inter-connected clusters and groups of individuals (e.g., a sister, mom, and boyfriend) may lead one to actively engage with those in a cluster in pursuit of a common goal or activity, in which case any given member (such as the ego) might more passively be drawn in, rather than actively initiating or planning the activity. Indeed, if such a network initiates more group activity with a depressed ego, then it is likely that there would be less need for egos to initiate depression mitigating behaviors by themselves, therefore, suggesting an unclear correlational pattern between the connectedness dimension of cohesiveness and depression mitigation.

Third, in our study, networks for depressed and non-depressed individuals did not differ in constraint. However, our results also showed that this structure—a more radial rather than interconnected confidant network—can be very important in motivating depression mitigation behaviors. For depressed individuals, a radial confidant network can very well represent participants’ actively seeking information and privacy management. Indeed, we found that networks with more constraints (i.e., when more alters are connected) can hinder depression mitigating behaviors. This result is consistent with previous studies that showed low constraint can be beneficial for individuals under other information-seeking circumstances (where access to unique types of information is desired [27]. Figure 1A is a relatively less constrained confidant network, the ego seems to be the bridge between the “therapist (as nick-named by the participant)” and the family members. This structure indicates that the depressed participants are more in control of his/her conditions. Compare this to the network shown in Figure 1B, where the two “shrinks (also as nick-named)” of the ego know each other, and also know the ego’s closest confidants, such as her boyfriend. The information flow between those connected alters may “constrain” the ego’s active management of his/her mental wellbeing. Indeed, within a more constrained confidant network where everyone knows ego’s state of depression, egos may see less reward value from taking active control of his/her condition. That is, in a “less cohesive network” with more structural holes, depressed egos may have more needs, since they may have less contact with alters and, therefore, have more motivation to initiate depression mitigating behaviors. Further studies are needed to investigate the mechanisms that may enable “less constrained” networks to play a positive role. For example, are such networks effective because they encourage those who are depressed to actively seek control of their mental health conditions?

Lastly, our results showed that when accounting for the effects of network structures (i.e., size, network composition), depression diagnosis (depressed or not) is not a significant predictor of depression mitigation behaviors for those who have a confidant network. Those results speak to the importance of having a confidant network. That is, depression mitigation behaviors can be activated, whether one is clinically depressed or not, via a confidant network. Indeed, as reported in previous sections, those who do not have a confidant network, also despite their depression diagnosis, showed significantly less depression mitigation behaviors. Those results indicate that depression may be less likely to occur if one has the right number of people to whom one can confide (represented by size), especially with this group of people organized in a way that can help the depressed to manage their symptoms (represented in constraint). Indeed, having the right type of confidant networks matters a lot in helping one combat depression. Perhaps subsequent interventions can use this result to develop more effective interventions in developing interpersonal connections.

## 6. Implications of Network Interventions

The implications of these results could be profound in designing depression interventions from a network perspective. The current work shows the importance of situating the clinically depressed individuals within a network of members who know and care about the ego’s situation (i.e., size), and meanwhile where the ego has the capacity to actively coordinate with each of these others to help him or her combat depression (e.g., ego actively managing depression condition). Especially, the development of social media platforms and mobile technologies can enable such interventions. For example, a just-in-the-moment, adaptive intervention (JITAI) approach [53], is a means of delivering interventions such that participants can receive in-the-moment feedback on their social network platforms or mobile phones; it has already been shown to be quite advantageous in improving mental health outcomes [54]. Adopting the JITAI framework, participants can get in-the-moment feedback both on support-seeking behaviors and on maintaining privacy. Therefore, JITAI interventions can have the capacity to (1) identify the social support network of participants on social network sites, (2) create messages/intervention materials in helping participants to better communicate while maintaining personal privacy when participants need help, and (3) monitor the evolution of social network structures among the participants. Such interventions may be able to elicit a change of social support structure for the depressed participants (especially altering constraint structures).

Indeed, the challenge might be to provide the ego with optimal support while navigating privacy and communication issues throughout this care network. However, if those challenges are addressed (as in our JITAI example), changing the social network structures among the depressed (e.g., encouraging the patient to actively bridge information between health providers and family members) could facilitate information flow regarding the patient [55]. It could provide caretakers of depressed individuals with greater levels of social support and generate synergetic effects in helping the depressed individuals [56,57]. Indeed, one’s social networks could be another promising targeted intervention component that matters in combating depression.

## 7. Limitations and Future Directions

Even though this study has shown the importance of confidant networks in depression, it has some limitations. Specifically, it is still limited in its scope in terms of the studied problem. First, because of the limits of the design, we were not able to collect changes in members of confidant networks over time. Therefore, results of this study can only be interpreted as associations rather than as providing evidence of a causal relationship between depression mitigating behaviors and changes in one’s social relationships. In a subsequent study, it would be even more meaningful to examine changes in network structures over time from the onset of depression to the mitigation of depression-related behaviors and recovery. Second, because of the voluntary nature of this study, most of our participants were female, therefore, the results of this study may not generalize to the social networks of men who are clinically depressed (or not). Subsequent analyses could take a further look at the role of gender by recruiting more male participants. Moreover, because ResearchMatch.org operates on a voluntary basis, our sample may be subjected to the same self-selection bias. There is a possibility that the results of the study cannot be generalized to those who are severely depressed or socially isolated. Indeed, in our analysis, we found that depressed participants reported greater engagement in depression mitigation behaviors than those who were not depressed. While this is a positive sign, it can also indicate that our depressed participants from researchmatch.org could be more active than other depressed individuals. Furthermore, as one astute reviewer noted, researchers have not included a measure of the quality (as well as the quantity) of support from social network members and this should also be considered in future work. Research on individuals with depression may require more rigorous observational studies than the self-reported methods we have used in this study. Third, due to the scope of the current paper, we were not able to examine other variables that can play a role in mitigating depression behaviors. For example, our data did not enable us to further look into how economic backgrounds, religion, and ethnicity affect depression mitigation. There is a need for subsequent research to examine further the role of those variables at different levels of scale.

## 8. Conclusions

Nonetheless, the results of our study have depicted a robust picture regarding the structures of clinically depressed individuals’ confidant networks. Agreeing with previous findings on the importance of larger social network size and wide compositions (diverse group members) in confronting depression [2,58], we also found other important structural features of networks matter. Indeed, our study suggested the importance of a network structure suggesting active confidant seeking (shown by low constraint) and that may be an important indicator of positive depression mitigating behavior processes and outcomes.

## Figures and Tables

**Figure 1 ijerph-18-08388-f001:**
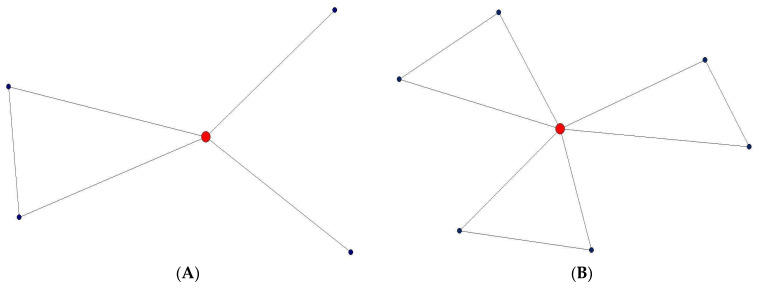
(**A**) A less constrained confidant network of a clinically depressed participant. (**B**) A more constrained confidant network of a clinically depressed participant.

**Table 1 ijerph-18-08388-t001:** Depression mitigation scores by clinical diagnosis.

	Clinically Depressed		Non-Depressed		*t* (1055)
	*M*	SD	*M*	SD	
There were certain things I needed to do that I didn’t do.	4.24	1.88	4.54	1.83	−2.53 **
I am content with the amount and types of things I did.	3.92	1.74	3.39	2.18	4.07 **
I engaged in many different activities.	4.10	1.87	3.74	2.17	2.75 *
I made good decisions about what type of activities and/or situations I put myself in.	4.47	1.69	3.85	1.84	5.42 *
I did things that were enjoyable.	4.77	1.60	3.94	1.83	7.51 *

** *p* < 0.01, * *p* < 0.05, * Participants rated on a 1 to 7 scale with 1 being not like me at all, 7 being a lot like me. We reverse-coded items that should be reverse-coded such that a score of 7 means the highest in depression mitigation behaviors.

**Table 2 ijerph-18-08388-t002:** Demographic information of participants: age, gender, income, ethnicity, and depression diagnosis.

	Clinically Depressed			Non-Depressed	
Age		39.05	13.59	37.85	11.92
Education (in years)		12.31	5.12	13.32	4.5
		N	Percentage	*n*	Percentage
Gender	Male	69	17.7	144	21
	Female	321	82.3	542	79
Income	Less than 30,000	106	27.2	180	26.2
	30,000–39,999	45	11.5	80	11.7
	40,000–49,999	47	12	89	13
	50,000–59,999	46	11.8	89	13
	60,000–69,999	27	7	48	7
	70,000–79,999	28	7.3	49	7.1
	80,000–89,999	23	5.9	37	5.4
	90,000–99,999	15	3.9	24	3.5
	100,000 or more	51	13.2	77	11.2
Ethnicity	White/Non-Hispanic	281	72	515	75
	Hispanic	38	9.8	56	8.1
	Black	39	10.1	55	8
	Pacific Islander	4	1.1	8	1.1
	Native American	2	0.6	4	0.6
	Middle Eastern	2	0.6	4	0.6
	Asian American	22	5.6	41	6
Depression diagnosis	Major Depressive Disorder	230	59		
	Persistent Depressive Disorder	59	15		
	Bipolar Disorder	35	9		
	Seasonal Affective Disorder	8	2		
	Psychotic Depression	1	0.2		
	Peripartum (Postpartum) Depression	5	1.2		
	‘Situational’ Depression	15	3.9		
	Atypical Depression	9	2.4		
	Mild depressive diagnosis	46	11.7		

Note: some participants may choose not to disclose or skip some of the survey questions, therefore, we may expect the numbers of participants in this table to be different from participants in the actual analysis.

**Table 3 ijerph-18-08388-t003:** Network structures by clinical diagnosis.

	Clinically Depressed		Non-Depressed		*t*	95% CI		Cohen’s d
	*M*	SD	*M*	SD		Upper	Lower	
Size (*n* = 1002)	4.15	1.71	4.19	1.63	−0.31	−0.25	0.17	0.05
Constraint (*n* = 824)	0.61	0.15	0.61	0.14	−0.26	−0.02	0.02	0.07
Mean Degree (*n* = 824)	2.23	1.12	1.95	1.12	−4.24 **	0.19	0.40	0.25
Gender (Index of Qualitative Variation, *n* = 824)	0.68	0.33	0.74	0.29	−3.16 **	0.24	0.10	0.19
Kin Proportion (*n* = 824)	0.29	0.38	0.29	0.34	−0.60	−0.03	0.51	0.06

** *p* < 0.01.

**Table 4 ijerph-18-08388-t004:** Hierarchical linear regression results for depression mitigation behaviors: evaluating the role of network size.

Variable		*B*	95% CI for *B*		*SE B*	*β*	*R* ^2^	Δ *R*^2^
			LL	UL				
Step1								
	Constant	4.15	3.88	4.48	0.15		0.00	0.00
	Kinship Proportion	0.05	−0.30	0.38	0.17			
	Gender IQV	0.18	−0.20	0.50	0.18	0.01		
	Clinical Diagnosis	0.07	−0.12	0.25	0.10	0.04		
Step2						0.03		
	Constant	3.66	3.19	4.13	0.24		0.01	0.01 *
	Kinship Proportion	−0.01	−0.35	0.33	0.17	0.00		
	Gender Diversity	0.17	−0.21	0.50	0.18	0.03		
	Clinical Diagnosis	0.08	−0.12	0.26	0.10	0.03		
	Network Size	0.11 **	0.03	0.19	0.04	0.09		

Note. CI = confidence interval; LL = lower Limit; UL = Upper Limit. SE = Standard Error, IQV: Index of Qualitative Variation, * *p* < 0.05, ** *p* < 0.001, *n* = 788 participants who have more than 3 confidants.

**Table 5 ijerph-18-08388-t005:** Hierarchical linear regression results for depression mitigation behaviors: evaluating the role of mean degree.

Variable		*B*	95% CI for *B*		*SE B*	*β*	*R* ^2^	Δ *R*^2^
			LL	UL				
Step1								
	Constant	4.15	3.84	4.46	0.15			
	Kinship Proportion	0.05	−0.28	0.37	0.16	0.01	0.00	0.00
	Gender IQV	0.18	−0.22	0.55	0.19	0.04		
	Clinical Diagnosis	0.07	−0.13	0.27	0.10	0.03		
Step2								
	Constant	4.25	3.82	4.62	0.20			
	Kinship Proportion	0.11	−0.24	0.47	0.18	0.02	0.00	0.00
	Gender Diversity	0.19	−0.21	0.55	0.19	0.04		
	Clinical Diagnosis	0.06	−0.14	0.25	0.10	0.02		
	Mean Degree	−0.05	−0.19	0.09	0.07	−0.03		

Note. CI = confidence interval; LL = lower Limit; UL = Upper Limit. SE = Standard Error, IQV: Index of Qualitative Variation, *n* = 788 participants who have more than 3 confidants.

**Table 6 ijerph-18-08388-t006:** Hierarchical linear regression results for depression mitigation behaviors: evaluating the role of constraint.

Variable		*B*	95% CI for *B*		*SE B*	*β*	*R* ^2^	Δ *R*^2^
			LL	UL				
Step1								
	Constant	4.15	3.84	4.46	0.16		0.00	0.00
	Kinship Proportion	0.05	−0.26	0.38	0.16	0.01		
	Gender IQV	0.18	−0.18	0.51	0.18	0.04		
	Clinical Diagnosis	0.07	−0.12	0.26	0.10	0.03		
Step2								
	Constant	4.82	4.21	5.42	0.30		0.01 *	0.01 *
	Kinship Proportion	0.01	−0.31	0.33	0.16	0.00		
	Gender Diversity	0.18	−0.18	0.51	0.18	0.04		
	Clinical Diagnosis	0.07	−0.12	0.26	0.10	0.03		
	Constraint	−1.15 *	−2.08	−0.19	0.48	−0.09		

Note. CI = confidence interval; LL = lower Limit; UL = Upper Limit. SE = Standard Error, IQV: Index of Qualitative Variation, * *p* < 0.05, *n* = 788 participants who have more than 3 confidants.

## Data Availability

Data from study are available upon request. A request for data can be made to the corresponding author: Liyuan Wang: liyuanwa@usc.edu.

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
