# Peer review of "The Structure of Social Support in Confidant Networks: Implications for Depression"

_ijerph, 2021, doi:10.3390/ijerph18168388_

Round 1

Reviewer 1 Report

Dear authors!

Thank you for the interesting study you provided. It was a pleasure to read it. From my opinion, you did a great job by collecting such a data and described the process in details. Moreover, the research question is relevant for the scope of the Journal and has a lot of potential interest to the readers both from academic and practical view. At the same time, the draft of the paper has some aspects needed to be improved before the production process:

  1. From my opinion, you should provide the research gap and contribution more clearly. Specifically, to answer in the Introduction - why this topic has the impact? Why we should study this phenomenon? For now, the whole paper looks more that description of the research process rather than the finished empirical study with some important findings.  I recommend including some references from academic journals and magazines/media to underline the importance of this issue. Furthermore, you can show some examples (or figures) how the depression and the networks can influence the mental health in communities/schools/regions/countries and connect this with economic indicators. Try to look at the findings in a way that is more general - it definitely will increase the contribution and expand the audience.
  2. I did not understand why you put one hypothesis and one research question. It seems that you need to be more consistent. Moreover, in the literature review I expected to see deeper explanation of the essence of the phenomenon you study. From my point of view, it is not very clear relationship between the theoretical background and conclusions for it and empirical part.
  3. Please, explain why you use HLM, provide all the necessary equations and describe the results. Also I would be glad to know more about the data rather than the process how you collected and verified it (I think it can be moved to the Appendix). What are the preliminary conclusions you can make looking at the descriptive stats and correlations?
  4. Please, in the results section provide the information about the hypothesis – whether it was proved as well as the answer for the research question.
  5. In the discussion section underline the contribution and position the paper among the other in the field.

Good luck!

Reviewer 2 Report

The topic is important. Nowadays growing number of people experience depression. The authors do not take in to consideration cultural differences among ethnic groups in US in reporting their health problems as well as created networks., influence of religions (e.g. difference between Protestants and Catholics)for readiness to look for support/help of other people.  The subject has been described in American literature several times.. Number of White participants included in the study allows to study the differences among them. 

Also age can differentiates types of networks. 

The paper as such can be considered as very introductory to the studied problem.

Reviewer 3 Report

Reviewer comments Manuscript 1243122

Thank you for the opportunity given to me to revise the manuscript “The Structure of social support in confidant networks: Implication for depression”. The study offers an original point of view to address a key point concerning strategies to cope with depression, that is the promotion of behaviours that can mitigate depression. The study appears to address with an important key point for the implementation of prevention and intervention strategies for depression by means of non-pharmacological approaches that can be implemented with non-responsive subjects or as complementary intervention. Unfortunately, the study presents many methodological limits and also inaccuracy in the presentation of both methods and results that impede the evaluation of the discussion and conclusions made by Authors. The reviewer conclusion is to reject the manuscript in the actual form.

Main points

The recruitment methods should be described with more precision especially on i) case definition. Did authors base case definition only by Researchmatch.org registry and control of the capacity to declare date of diagnosis by the patients? Did Authors verify the depression condition/disorders by means of second evaluation step or the Researchmatch.org registry foresees reliable control of the patients’ declarations? The description of the study design failed to define exclusion criteria. Did Authors control for the presence of other disorders other than mental ones, i.e chronic conditions, in people defined clinically without depression? Table 1 shows that clinically not-depressed people had lower scores than depressed people and this data should be discussed and a more in deep analysis of the characteristics of no-depressed people undergone. Interpretation of results is clearly influenced by the condition of the population used for comparison with depressed people. In Table 1 one would expect the exact opposite for means reported in the first and third column.

Authors said that they used “the five items reported in Table 1 in a composite variable representing overall depression mitigation engagement”. However, Authors did not define better this composite variable (by means of the formula) and consequently the Results section it is hard to understand. When Authors compare depressed vs non depressed people by the effects of independent variables, which is the dependent variable that they used? How Authors yielded the composite scoring considered the dependent variable? Which is the range of variation of this variable? Interpretation of linear regression results well depend on this range of variation.

Furthermore, many inconsistencies were detected in the manuscript: 1) total sample use for the analysis. Table 2 report a total different from the total stated i) in methods 3.1 section, ii) in Table 1 and iii) in the result section 4.2; 2) Many inconsistences are present between text and tables’ content (i.e. authors described in the text the likert scale used to measure BAD as a seven-points scale ranging from 1 (not at all) to 7 (completely). However, in Table 1, Authors described the same likert scale as “1 being not like me at all” and “7 being a lot like me”. In Table 3 are reported figures different from that those reported in the description of results); 3) Overall, titles of the tables should include basic and independent from text information useful to drive the reading. A new title should be done for Table 1 and ff. Table 2 would be improved by the inclusion of column titles and also the total n for depressed and non-depressed people. Table 4, 5, 7 use different definition for the same variable inside; Where is Table 6? 4) Measure of social network need a clearer definition (i.e. difference between network size and mean degree: how number of ties are collected? gender diversity: which is the formula to yield it?).

Author Response

Please see attchement

Reviewer 4 Report

Dear Authors

I apprciate the text as reffering to the ineteresting and good quality research. Still I have some issues that may help to improve the text:

For introduction - I will also put some theory against the reasoning presented - some depressed individuals are surrounded by people that have strong bonds but still feel very lonely and sometimes having one supporter with a good quality of support works perfectly. This I will also use for interpretation of the results.

H1 - the wording may be misleading - as the network size and cohesion are the terms that a little bit overlap here. This definitely needs clarification.

Table 1 - according to the data clinically depressed use better everyday coping (mitigation) strategies than those non-depressed. Is it really so or there is a mistake in the table? If not, in my opinion this result is theoretically unexpected and should be discussed.

Gender index - do I understand correctly that 0 does not distinguish whether those same sex supporters are all men or women? It certainly would make a difference depending of the gender of an individual that receives support.

In limitations I suggest to write more about the volunteer samples that are different to general population in many aspects such as a level of education, some psychological characteristics, etc.
